# Phosphorylated Proteins from Serum: A Promising Potential Diagnostic Biomarker of Cancer

**DOI:** 10.3390/ijms232012359

**Published:** 2022-10-15

**Authors:** Rishila Ghosh, Rakin Ahmed, Hafiz Ahmed, Bishnu P. Chatterjee

**Affiliations:** 1Department of Oncogene Regulation, Chittaranjan National Cancer Institute, Kolkata 700026, India; 2GlycoMantra Inc., Biotechnology Center, University of Maryland Baltimore County, Baltimore, MD 21227, USA

**Keywords:** biomarker, phosphoproteins, phosphorylation, dephosphorylation, cancer, serum

## Abstract

Cancer is a fatal disease worldwide. Each year ten million people are diagnosed around the world, and more than half of patients eventually die from it in many countries. A majority of cancer remains asymptomatic in the earlier stages, with specific symptoms appearing in the advanced stages when the chances of adequate treatment are low. Cancer screening is generally executed by different imaging techniques like ultrasonography (USG), mammography, CT-scan, and magnetic resonance imaging (MRI). Imaging techniques, however, fail to distinguish between cancerous and non-cancerous cells for early diagnosis. To confirm the imaging result, solid and liquid biopsies are done which have certain limitations such as invasive (in case of solid biopsy) or missed early diagnosis due to extremely low concentrations of circulating tumor DNA (in case of liquid biopsy). Therefore, it is essential to detect certain biomarkers by a noninvasive approach. One approach is a proteomic or glycoproteomic study which mostly identifies proteins and glycoproteins present in tissues and serum. Some of these studies are approved by the Food and Drug Administration (FDA). Another non-expensive and comparatively easier method to detect glycoprotein biomarkers is by ELISA, which uses lectins of diverse specificities. Several of the FDA approved proteins used as cancer biomarkers do not show optimal sensitivities for precise diagnosis of the diseases. In this regard, expression of phosphoproteins is associated with a more specific stage of a particular disease with high sensitivity and specificity. In this review, we discuss the expression of different serum phosphoproteins in various cancers. These phosphoproteins are detected either by phosphoprotein enrichment by immunoprecipitation using phosphospecific antibody and metal oxide affinity chromatography followed by LC-MS/MS or by 2D gel electrophoresis followed by MALDI-ToF/MS analysis. The updated knowledge on phosphorylated proteins in clinical samples from various cancer patients would help to develop these serum phophoproteins as potential diagnostic/prognostic biomarkers of cancer.

## 1. Introduction

Phosphorylation is one of the post-translational modifications (PTMs) of protein in which a phosphate group (PO_4_^−^) is chemically attached to specific amino acid residues of a protein. Phosphorylation occurs through catalyzation of a kinase enzyme by adenosine triphosphate (ATP), a phosphate anhydride, which acts as a donor of a phosphate group [1]. More than one-third of the protein phosphorylation (O-phosphorylation) occurs on serine (Ser or S), threonine (Thr or T), and tyrosine residues (Tyr or Y) [2]. Specifically, serine accounts for 86.4% of protein phosphorylation, followed by 11.8% for threonine and only 1.8% for tyrosine [3,4]. The phosphorylation/dephosphorylation of protein is rigidly controlled by the interplay of two types of enzymes: protein kinases and phosphatases [5]. Almost all the cellular and biochemical signaling pathways are activated and deactivated by either phosphorylation or dephosphorylation [6]. Recently, it has been reported that aberration of phosphorylation is associated with various pathophysiological processes [7,8]. Frequent aberration of phosphoproteins was observed in biological samples like tissue, serum, saliva, and urine of the patients of different diseases, including cancer [9]. In addition, different expression of phosphoproteins was reported in serum of chronic hepatitis B (CHB), liver cirrhosis (LC), and hepatocellular carcinoma (HCC) patients [10].

Cancer is one of the most complex and aggressive diseases present among the global population, including India. Despite profound advancement in diagnosis and treatment modalities, many of the cancers still show a high relapse rate with guarded prognosis, leading to significant morbidity and mortality [11]. Thus, effective, novel, and precise biomarkers are necessary to aid in early diagnosis and treatment of cancer in order to increase the patient rate of survival. Biomarkers indicate a change in expression of biomolecules, which are associated with susceptibility and progression of a particular disease such as cancer [12]. In order to use biomarkers for diagnosis, clinical samples such as tissue, blood, saliva, urine, and other biological fluids are needed. In spite of characterization of a large number of changes in genetic/epigenetic/protein expression associated with cancer, it is logical to target proteins in clinical research as biomarkers because the proteins are direct participants of oncogenesis [13,14]. In addition, detection and quantification of proteins and their post-transnationally modified forms are easily accessible by different immunological methods [15]. Thus, targeting phosphorylated serum proteins may be effective biomarkers for cancer patients [16]. In this review, phosphorylated proteins in clinical samples from various cancer patients are discussed and this updated knowledge would help to develop these serum phophoproteins as potential diagnostic/prognostic biomarkers of cancer.

## 2. Aberrant Phosphorylation in Cancer

Protein phosphorylation is a vital step for the coordination of cellular and molecular functions, such as the regulation of metabolism, proliferation, apoptosis, subcellular trafficking, inflammation, and other important physiological processes. Thus, altered expression of phosphoprotein will lead to deregulation of different signaling pathways as well as development and progression of cancer. Different expression patterns of various phosphoproteins have been reported in sera of various cancers and are discussed in this review.

### 2.1. Hepatocellular Carcinoma

Hepatocellular carcinoma (HCC) is the sixth most commonly diagnosed malignancy worldwide, and the third most common cause of cancer-related death. HCC is associated with multiple risk factors and cofactors. In the majority (80–90%) of patients, HCC is preceded by cirrhosis. In this regard, chronic hepatitis B virus (HBV) or hepatitis C virus (HCV) infection is of particular concern. HBV-related chronic hepatitis is responsible for an estimated 50–80% of HCC cases worldwide, whereas 10–25% of cases may be due to HCV infection [17].

Despite existing serum diagnostic markers of HCC such as Alpha-fetoprotein (AFP), scientists have currently concentrated on characterization of phosphoprotein profiles in the serum of HCC patients to develop a precise early diagnostic biomarker of HCC. Hu et al. conducted a phosphoproteomics study on serum of 12 HCC patients and 12 healthy individuals [18]. Prior to MS analysis, phosphopeptide enrichment was performed using TiO_2_ immobilized mesoporous silica particles. MALDI-TOF MS analysis initially identified four phosphopeptide peaks, indicating a potential difference between HCC patients and healthy individuals. The MS-based quantification of phosphopeptide, followed by a partial least-squares discriminate analysis (PLS-DA), revealed that the peptide D[pS]GEGDFLAEGGGV was upregulated, whereas the peptide of AD[pS]GEGDFLAEGGGVR was downregulated greatly. Minimal change was observed for the other two peptides, AD[pS]GEGDFLAEGGGV and D[pS]GEGDFLAEGGGVR. The result indicates that the first two phosphopeptides, D[pS]GEGDFLAEGGGV and AD[pS]GEGDFLAEGGGVR, may be effective diagnostic biomarkers for liver cancer [18]. There are several biomarkers that have the potential to dramatically improve the early detection of HCC. We have listed some serum protein biomarker in Table 1.

### 2.2. Gallbladder Cancer

Gallbladder cancer (GBC) is a distinct type of biliary tract cancer that is rare, aggressive, and has limited treatment options apart from surgical resection, which has an estimated 5 year survival rate of 2% in metastatic disease. In 2022, an estimated 12,130 GBC and other biliary cancers are expected to be diagnosed, with an estimated 4400 patients dying from these diseases. Prognosis is particularly poor in elderly and racial minorities [24]. Tan et al. identified 24 differentially expressed proteins, which includes 12 upregulated and 12 downregulated proteins between gallbladder cancer patients and healthy controls [25]. Haptoglobin, S100A10, and other identified proteins may be potential molecular targets for early diagnostic and therapeutic application of GBC [25]. The immunohistochemistry showed that high expression of phosphorylated extracellular signal-regulated kinase 1(ERK1) at 202 threonine and ERK2 at 204 threonine residues in a gallbladder tumor was significantly associated with poor survival of gallbladder cancer patients [26].

### 2.3. Cholangiocarcinoma

Cholangiocarcinoma (CCA) is a highly malignant tumor arising from the epithelial cells lining the bile duct. The five-year survival rate for extrahepatic and intrahepatic bile duct cancer is 10% and 9%, respectively, whereas if they are diagnosed at an early stage, the 5-year survival rates are 17% and 25%. The slow progression makes it difficult for early diagnosis, and most cases are detected in advanced stages. Most of the approaches regarding diagnosis of CCA have focused on analysis of tumor tissue, biopsy, and proteomics of serum samples. In one of those studies, proteins FAM19A5, MAGED4B, KIAA0321, RBAK, and UPF3 were identified in the serum through a proteomic approach using highly stringent analysis with cross-validation [27]. The presence of these proteins can potentially discriminate patients with CCA from individuals having benign biliary tract diseases (BBTD). Kotawong et al. took a precise diagnostic approach for CCA by implementing phosphoproteomics on serum of 10 CCA patients, 5 *Opisthorchis viverrini* (OV) infected patients, and 5 healthy individuals [28]. In this study, 2D gel electrophoresis characterized 300 spots as phosphoproteins with two prominent 21 kDa upregulated spots, 21A and 21B. The LC-MS/MS analysis of the 21A and 21B spots revealed 98 and 64 identified proteins, respectively. Among these, MeV (Multiple Experiment Viewer) program-based bioinformatics and statistical analysis identified two proteins (trafficking protein particle complex subunit 5 and p115) which were significantly upregulated in plasma of CCA patients compared to the non CCA group. Those two above proteins are potential diagnostic biomarkers of CCA by combination of kinetic Monte Carlo (KMC) with a statistical *t*-test analysis [28].

### 2.4. Gastric Cancer

Gastric carcinoma (GC) is the fifth most common cancer and the fourth leading cause of cancer-related death worldwide in 2020. In the same year, an estimated 1.1 million cases (720,000 males and 370,000 females) of GC were diagnosed worldwide [29]. The serum phosphoproteomics study was also executed in GC patient samples to overcome the non-specific diagnosis of gastric cancer with existing carcinoembryonic antigen (CEA) and carbohydrate antigen 19-9 (CA19-9) biomarkers. In order to understand the phosphopeptide profile in patients’ serum, Zhai et al. initially conducted a phosphoproteomics profile on 20 GC patients and 20 healthy individuals by MALDI-TOF MS analysis [30]. After phosphopeptide enrichment with TiO_2_, the mass spectrum of the serum sample initially showed four remarkable ion peaks. Subsequently, tandem mass spectrometry (MS/MS) characterized four ion peaks corresponding to phosphopeptides, ADpSGEGDFLAEGGGV (F1), DpSGEGDFLAEGGGV (F2), DpSGEGDFLAEGGGVR(F3) and ADpSGEGDFLAEGGGVR (F4) at m/z 1389.3, 1460.4, 1545.5, and 1616.6, respectively, which were all derived from fibrinopeptides A. Further, absolute quantification of the four endogenous phosphopeptides in serum by MS revealed the F3 phosphopeptide was significantly downregulated in GC samples compared to healthy controls. They extended their study on a training set of 40 GC patients and 30 healthy individuals, and a validation set of 20 GC patients and 30 healthy individuals, in order to establish serum phosphopeptides as potential diagnostic biomarkers of GC [31]. Interestingly, the F3 level in sera was significantly reduced in eight out of nine patients (89%) at stage I. When F3 was incorporated into receiver operating characteristic (ROC) analysis independently, GC could be distinguished from controls with 88.3% specificity and 96.7% sensitivity. This suggested evaluation of F3 phosphopeptide level from fibrinopeptides A of fibrinogen in serum of GC patients could be used as potential early diagnostic marker of GC [31].

### 2.5. Lung Cancer

Lung cancer (LC) is the most common malignancy worldwide and is the leading cause of death. There were more than 2.2 million new cases of lung cancer in 2020. In India, lung cancer constitutes 6.9% of all new cancer cases, and lung cancer-related death accounts for 9.3% of all cancer-related death. Small cell lung cancer (SCLC) accounts for <20% of LC cases, whereas non-small cell lung carcinoma (NSCLC) cases are a majority, with adenocarcinoma (ADC) 32%, squamous cell carcinoma (SCC) 30%, and large cell carcinoma (LCC) 10%. Despite the progress in LC research and advancement in treatment strategies, the five-year survival rate for patients with LC was <15%. Poor prognosis is mainly attributed to late diagnosis, with the majority of LC patients diagnosed at an advanced stage when the surgical resection is hardly possible. Tremendous amounts of evidence suggest that genetic abnormalities contribute to the development of lung cancer. These molecular abnormalities may serve as diagnostic, prognostic, and predictive biomarkers for this deadly disease.

Cyclase-associated protein 1 (CAP1) has been shown to overexpress in most cancer types and its high expression is correlated with poor prognosis [32]. Recent results showed that CAPI was overexpressed in NSCLC serum samples when compared with the healthy control group and correlated with poor clinical outcomes [32]. The role of CAPI in lung cancer progression has been implicated to two tandem phosphorylation sites, S308 and S310. In both in vitro and in vivo experiments, the phosphorylated S308 and S310 in CAP1 promoted lung cancer cell proliferation, migration, and metastasis [32]. However, de-phosphorylated S308 and S310 in CAP1 inhibited the proliferation ability in A549 cells [32].

### 2.6. Prostate Cancer

Prostate Cancer (PCa) is the second most commonly diagnosed cancer, and the fifth leading cause of cancer-related death among men worldwide, with an estimated 1,414,000 new cancer cases and 375,304 deaths in 2020 [33]. The serum prostate specific antigen (PSA) test has been the leading method of screening for prostate cancer. The PSA test, in conjunction with other common tests like digital rectal examination (DRE) or transrectal ultrasound (TRUS), can reveal the probability of the incidence of prostate cancer [34]. Based on the report from the National Prostate Cancer Detection Project of the American Cancer Society, 92% of cancers detected by PSA, DRE, and TRUS from an annual testing are localized to the prostate [35]. However, the treatment options for all localized prostate cancer are not the same, as some prostate tumors are aggressive and some are indolent. Unfortunately, the current clinical biomarkers for prostate cancer are not ideal to specifically distinguish between those patients who should be treated adequately to stop the aggressive form of the disease and those who should avoid overtreatment of the indolent form. To search for the reliable biomarkers of prostate cancer, Liu et al. assessed 80 serum samples from four groups: men without PCa, patients with low risk primary PCa, patients with high risk primary PCa, and patients with metastatic PCa (n = 20 per group) [36]. Among the target proteins, nine of them (PTN, MK, PVRL4, EPHA2, TFPI-2, hK11, SYND1, ANGPT2, and hK14) were found to be significantly increased in the metastatic PCa group compared to others. In another study, six (CASP8, MSLN, FGFBP1, ICOSLG, TIE2, and S100A4) out of 174 target proteins were found to be significantly decreased after radical prostatectomy (RP) in patient-matched serum samples from ten men with high grade and high-volume prostate cancer [37]. Tony et al. identified and evaluated a potential serum protein signature of disease recurrence in a cohort of PCa patients that received treatment with combined hormone and radiation therapy (CHRT) [38]. Label-free LC-MS/MS-based protein discovery on depleted serum samples from CHRT patients (n = 3) with disease recurrence and time-matched patient controls (n = 3) resulted in a total of 104 proteins that showed a significant change between these two groups. Using multiple reaction monitoring (MRM) assays, a panel of 41 putative prostate cancer biomarkers were then selected for evaluation in longitudinal serum analysis (Table 2). Thomas et al. described the development of multiplexed targeted mass spectrometry assays to quantify N-linked glycosite-containing peptides in serum using parallel reaction monitoring (PRM) [39]. They showed that a total of 41 of 43 previously identified *N*-linked glycosite-containing peptides were reproducibly quantified, with four proteins showing differential significance in serum from nonaggressive vs. aggressive prostate cancer patient serum (see Table 2).

### 2.7. Breast Cancer

Several studies reported the association of altered expression of serum proteins and changes in their glycosylation pattern and miRNA with the pathogenesis of breast cancer. These studies were conducted to find biomarkers for early diagnosis of the disease [40,41]. However, the potential use of serum phosphoproteins as biomarkers for diagnosis and prognosis of breast cancer was limited. Chen et al. [42] isolated and identified phosphoproteins in extracellular vesicles (EV) from human plasma as potential biomarkers to discriminate breast cancer patients from a healthy control. Eighteen breast cancer patients and six healthy individuals were included in their study. Phosphopeptides generated from the EV were enriched and analyzed by LC-MS/MS. They quantified 3607 and 461 unique phosphosites and identified 156 and 271 phosphosites with significant changes in microvesicles and exosomes, respectively. They compared these phosphosites representing 197 unique phosphopeptides that showed a significant increase in patients with breast cancer with all identified unique phosphopeptides in EV phosphoproteomes. They also found that a significant portion of these 197 phosphopeptides (>60%) were also identified by the proteogenomic study, which indicates that EV phosphoproteome was sensitive and could be used to identify phosphorylation events that were disease-specific [42]. In another study, protein profiling using antibody microarrays with 215 highly specific pre-selected antibodies was designed for different proteins and specific phosphorylation sites. Results indicate that p-S259 JunB, p-S79 JunB, and p-T512 ICAM-1 phosphoproteins were significantly upregulated in the plasma of breast cancer patients compared to healthy individuals [43]. This predicts that phosphoproteins could be a candidate for early diagnosis of breast cancer marker.

### 2.8. Colorectal Cancer

Annually, around 10% of cancer-related deaths are due to colorectal cancer (CRC) [44,45,46]. It has been observed that the incidence rates are rising in developing countries [44]. Males are at a higher risk for developing colorectal carcinoma compared to females. In 2020, the global CRC incidence rate in men (23.4 cases per 100,000 persons) was 44% higher than that in women (16.2 cases per 100,000 persons) [44].

Attempts are being continued to develop the early diagnostic marker of CRC with diverse approaches. Extracellular vesicle (EV) derived proteomic, in combination with phosphoproteomic of serum of CRC patients, preliminarily identified four proteins (FGA, FN1, S100A9, HP), with the phosphorylated forms significantly upregulated in samples from cancer patients compared to healthy individuals. DIA-MS analysis on independent cohort validated the results, showing that phosphorylation level of FGA in EV was significantly high in the serum of cancer patients. Further, a sensitivity test confirmed that FGA and its five phosphorylated forms distinguish between CRC and healthy individuals when compared to conventional diagnostic markers CA19-9 and CEA [47]. Thus, phosphosproteins could be a very sensitive early diagnostic marker for colorectal cancer.

### 2.9. Pancreatic Cancer

Pancreatic cancer (PaCa) ranks fourteenth among all the cancers worldwide as per GLOBOCAN 2018, with varying incidence rates across the globe. In India, there is also regional variation in the incidence across the country, with the northeast showing the highest rates [48]. Like cholangiocarcinoma, the five-year disease-free survival rate of PaCa is very low (~5%), as there was no sensitive and specific early diagnostic biomarker [49]. Some studies suggested a few serous proteins as specific biomarkers of PaCa, but translating these results into cost-effective and reliable clinical tests is very difficult [50,51]. Takano et al. attempted to observe the expression of circulating serum phosphoproteins in serum from 26 PaCa patients and 25 healthy individuals training set followed by validation set 1 and 2 to establish the impact of serum phosphoprotein as early diagnostic markers of PaCa. In this study, a Bio-Plex immunoassay was performed and revealed six phosphoproteins which showed significantly increased expression in serum from PaCa patients compared to healthy individuals. Among these, p-T202 ERK1, p-S201 ERK2, and p-T44 MEK1 proteins primarily proposed as potential diagnostic markers of PaCa. These phosphoproteins were found to be correlated with serum and tissue biopsy. However, the sensitivity of serum p-ERK1 and p-ERK2 was found to be better for prediction of stage-I PaCa than CA19-9, indicating that serum p-ERK1 and ERK2 could be a potential early diagnostic marker of PaCa [52].

### 2.10. Renal Cell Carcinoma

The highest prevalence of renal cell carcinoma (RCC) was accounted in the western countries and approximately accounts for 3% of all types of cancers. RCC patients without any metastasis showed a 93% five-year survival rate. However, with diagnosis at the advanced stage, the 5-year survival rate is 71% and is reduced to 14% when metastasis occurs. Ljungberg et al. reported a 2% annual increase in occurrence of RCC worldwide [53].

Diagnosis of RCC by biopsy is an invasive, painful, and complex method that often requires an expert radiologist in CT or USG. Instead of an invasive kidney biopsy, a simpler and non-invasive method such as using different miRNA, proteins, and phosphoproteins as biomarkers of kidney cancer has been considered. Recently, a phosphoproteomics study was conducted to distinguish kidney cancer accurately from non-cancerous conditions. Phosphoproteins extracted from extracellular vesicles and isolated from serum samples of chronic kidney disease (CKD), RCC patients, and from healthy controls were analyzed by LC-MS. LC-MS investigation with phosphopeptide enriched sample revealed 146 phosphoproteins which showed significant change in kidney cancer samples compared to the control. Similarly, 156 phosphoproteins were characterized, which showed a sharp change in CKD samples compared to control. Comparison between RCC and CKD samples revealed 44 phosphoproteins that were significantly different between these groups. Statistical analyses in the same set of samples revealed p-S107, S185 CRK-like protein (CRKL), and p-S298, S426 LYRIC (MTDH) phosphoproteins, which were found to be significantly high in abundance in RCC samples compared to CKD and control. However, concentration of apolipoprotein A-IV (APOA4) was found to be specifically increased in CKD samples compared to those of RCC and healthy control samples. Thus, these phosphoproteins could be proposed as potential diagnostic biomarker of kidney cancer [54]. Several new candidate proteins like CD14, MPO, NCF2, SOD2, and PARP1 were found to be upregulated and another set of proteins, MUT, ACADM, and PCK1, were downregulated in RCC. These proteins may be recognized as new biomarkers for RCC [55].

## 3. Currently Used Biomarker of Cancer

Cancer, when diagnosed at its earliest stage, has a chance of better treatment and survival rate [56]. However, the majority of cancers can remain asymptomatic until the appearance of specific symptoms in the advanced stage, except for certain primary cancer sites [57]. Thus, different types of cancer screening program are scheduled to increase the diagnosis of certain cancers at an early stage [58]. Cancer screening is generally executed by using different imaging techniques like USG, mammography, or MRI. Although imaging techniques are useful in detecting mass and size lesions, these are unable to distinguish between cancerous and non-cancerous cells for early diagnosis. To confirm the imaging result, solid and liquid biopsies were done for different cancer biomarker tests by proteomic study in the majority of the cancer [59]. The majority of the biomarkers currently used in clinical practice are proteins, some of them approved by the FDA (Table 3). Among them, some of the biomarkers are serum proteins, which were detected by ELISA. Although the FDA has approved several proteins for use as biomarkers of cancer, these proteins do not produce suboptimal sensitivities for precise diagnosis of the diseases. The detection of total protein in serum has some limitations due to one protein being associated with more than one clinical symptom and antibodies of total proteins having less specificity. However, expression of phosphoproteins is associated with a more specific stage of the particular disease. In addition, phospho-antibodies have high sensitivity and specificity compared to total proteins.

## 4. Detection of Protein Phosphorylation

There are many methods of phosphoproteins detection or analysis. One is phosphoproteins enrichment by (i) immunoprecipitation using phosphoserine/phosphothreonine/phosphor-tyrosine antibody, (ii) affinity chromatography by (a) metal oxide like TiO_2_, Ge_2_O_3_, Al_2_O_3_ (b) immobilized metal affinity chromatography, and (iii) ion exchange (strong anion exchange/cation exchange) chromatography (Figure 1). Another method is gel electrophoresis, followed by phospho-specific stain (ProQ diamond) and western blotting by phospho-specific antibody. Another method is ^32^P labelling gel electrophoresis or Edman degradation. These methods are still valid today and are used routinely in studying phosphoproteomics. The above methods are coupled to MS analysis because of their sensitivity, versatility, and reproducibility. Another approach is separation of phosphoproteins by 2D gel electrophoresis and an additional enrichment procedure such as phospho-specific staining, IMAC or immune-affinity followed by MS analysis to identify phosphoproteins present in cellular extracts. The immune-affinity method towards enrichment is an important approach for the identification of phosphotyrosine residues, as they are less abundant than phosphoserine and phosphothreonine residues.

## 5. Identification of Phosphoproteins as Cancer Biomarkers

The impact of biomarkers on cancer therapy can be considered a three-faceted approach involving prognostic, predictive, and diagnostic values. Prognostic biomarkers indicate the severity of the tumor and overall logical outcome for the patient, regardless of therapy, while a predictive biomarker provides information related to the effect of a therapeutic intervention. Prognostic type biomarkers are investigated through DNA or gene-expression signatures in the form of microarray. Diagnostic biomarker refers to a biological parameter that aids in the diagnosis of a disease and may serve in determining the disease progression and/or success of treatment. A laboratory, radiological, genetic anatomical, physiological, or other finding may help to differentiate one disease from another. It is important for stakeholders to perform a cost-effectiveness analysis when considering a new potential biomarker in order to determine whether it will make economic sense from the payer’s and society’s perspective.

## 6. Future Prospective of Serum Phosphoprotein as Biomarker

The change in expression of serum phosphoprotein in cancer patients could be an important biomarker, as delineated. As the current standard phosphoprotein technologies require expensive equipment and skilled operators, their use in routine clinical laboratories is not yet feasible. Therefore, further work is still needed to enhance the performance, reproducibility, and sensitivity of phosphoprotein detection methods in a manner where they can be routinely used in the clinical laboratory. Prior to clinical trial, it is necessary to characterize and validate the strong association between specific diseases with particular phosphoproteins. In addition, standardization of sensitivity and specificity of antibodies routinely used in clinical oncology is required for the point of care to large patients’ cohort with existing conventional biomarkers. With these issues solved, serum phosphoproteins could be an early and cost-effective diagnostic tool for cancer in the near future.

## Figures and Tables

**Figure 1 ijms-23-12359-f001:**
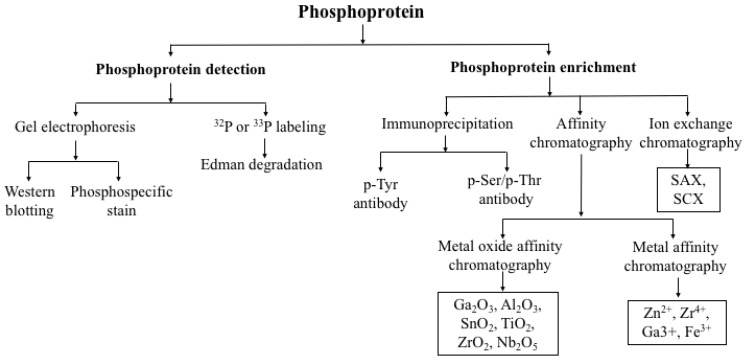
Detection of phosphoproteins.

**Table 1 ijms-23-12359-t001:** Disease marker and number of samples in HCC.

Name of Biomarker	Sample Size	Sensitivity	Specificity	Reference
AFP	836	53 (46–59)%	90 (87–93)%	[19]
AFP-L3	110	28 (22–34)%	97 (93–100)%	[20]
GOLPH 73	352	78.34%	77.59%	[21]
GPC-3	30	55.1 (47.9–66.2)%	97.0 (95.2–98.2)%	[21]
DKK1	831	41%-74%	87%	[22]
OPN	131	75 (58–93)%	67 (51–73)%	[23]

**Table 2 ijms-23-12359-t002:** List of MS-based targeted proteomics studies of PCa protein markers from serum.

Potential Biomarker	Sample Cohort	Source	Method	Ref.
C1QB, APOA4, CO9, ANT3, VTDB, PLMN, GPX3, ITIH4, CFAI, APOH, VTNC, IBP3, CLUS, APOA2, PEDF, TETN, CD 14, LG3BP, CFAH, FCN3, HPT, CO3, APOA1, APOC3, SAMP, HEMO, CO6, KLK3/PSA, A2MG, A1At, APOE, A2G1, TTHY, C1S, ZAG, AMBP, KNG1, CO4A, AACT, CAV1, TRFE	86 time-point samples from 3 PCa patients and 3 controls.	Immunodepleted serum	SRM-MS analysis of 59 peptides corresponding to 41 target proteins.	[38]
ITIH2, CD44, IGHG2, CDH13	25 aggressive PCa, 25 non-aggressive PCa	Serum	PRM-MS analysis of 41 *N*-glycosite-containing peptides corresponding to 37 target proteins	[39]

**Table 3 ijms-23-12359-t003:** List of conventional serum-based biomarkers for various cancers.

Cancer Type	Biomarker	Clinical Use	Methods	Ref.
HCC	AFP-L3 (Alpha Fetoprotein isoform 3)	Risk assessment for development of disease	Immunoassay	[60]
	Des-γ-carboxyprothrombin	Diagnosis	Immunoassay	[61]
	Glypican-3	Diagnosis	Immunoassay	[62]
Gallbladder cancer	Cancer antigen 19-9 (CA19-9)	Diagnosis	ELISA	[63]
Cholongiocarcinoma	Carbohydrate antigen-S27 (CA-S27)	Diagnosis and Prognosis	ELISA	[64]
	Angpt-2	Diagnosis	ELISA	[65]
Gastric cancer	Interferon Gamma Receptor 1 (IFNGR1), Notch Receptor 3 (Notch-3) TNF Receptor Superfamily Number 19-like (TNFRSF19L), Folate Receptor Beta (FR-beta), and SLAM Family Member 8 (SLAMF8)	Diagnosis	ELISA	[66]
Lung cancer	Progastrin-releasing peptide (ProGRP) Carcinoembryonic antigen (CEA)	Early Diagnosis	Immunoassay	[67]
Prostate cancer	Pro-prostate specific antigen (Pro-PSA)	Diagnosis	Immunoassay	[68]
	Total prostate specific antigen (Total-PSA)	Diagnosis	Immunoassay	[69]
Breast cancer	Circulating Tumor Cells (EpCAM, CD-45, Cytokeratins 8,18+, 19+), CA15-3	Prognosis	Immuno-magnetic capture/Immuno-fluorescence	[70,71]
Pancreatic cancer	S100 Calcium binding protein A4 (S100A4), S100 Calcium binding protein A 8 (S100A8), Cancer antigen 1 (CA1), Annexin V	Diagnosis	ELISA	[72]
RCC/Kidney cancer	Tumor necrosis factor receptor-associated factor-1 (TRAF-1)	Diagnosis	ELISA	[73]
	Heat shock protein 27 (Hsp 27)	Diagnosis	ELISA	[74]
	Human serum amyloid A (SAA)	Diagnosis	ELISA	[75]

## Data Availability

Not applicable.

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
