# Peer review of "Phosphorylated Proteins from Serum: A Promising Potential Diagnostic Biomarker of Cancer"

_ijms, 2022, doi:10.3390/ijms232012359_

Round 1

Reviewer 1 Report

Reviewer 1

Dear authors: Rishila Ghosh, Hafiz Ahmed, Bishnu P. Chatterjee. After an exhaustive revision, the manuscript (Manuscript ID ijms-1958591) is Accept after minor revision.

The manuscript by Rishila Ghosh et al. has investigated Phosphorylated proteins from serum: A promising potential diagnostic biomarker of cancer. Cancer is a fatal disease worldwide. Majority of the cancer remains asymptomatic at its earlier stage and the specific symptoms appear at the advance stage when the chances of treatment are not adequate. In this review, we have discussed the expression of different serum phosphoproteins in various cancers detected by phosphoprotein enrichment by immunoprecipitation using phosphospecific antibody and affinity chromatography on metal oxide followed by LC-MS/MS or by 2D gel electrophoresis followed by MALDI-ToF/MS analysis.

In general, the topics of the manuscript is closely connected to the journal's objectives. The described studies is very interesting. The abstract is good. The introduction is concise and precise, and the manuscropt has updated references until 2022.

In general the manuscript contain relevant paragraphs that have been discussed.

After close evaluation of the paper I suggest revision according to the next points:

1. The some subsections need to an important improve (2.5. Lung cancer, 2.6. Prostate cancer), since these subsections are very poor.

2. Figure 1 "Detection of phosphoproteins." needs to be improved.

3. Table 1 "Disease marker and number of samples in HCC." needs to be improved.

4. Indicate the element of novelty.

Author Response

We thank the reviewers for their constructive criticisms and evaluations of the manuscript. Their comments and suggestions significantly contributed to strengthen this revised manuscript. In the revised manuscript, every effort has been made to incorporate all suggestions made by the reviewers, and address their concerns. All changes/updates in the manuscript were underlined. 

"In general, the topics of the manuscript is closely connected to the journal's objectives. The described studies are very interesting. The abstract is good. The introduction is concise and precise, and the manuscript has updated references until 2022. In general, the manuscript contains relevant paragraphs that have been discussed."

Response: We thank the reviewer for finding our manuscript suitable for the journal.

Comment 1: Some subsections need improvement (2.5 Lung cancer, 2.6. Prostate cancer), since these subsections are very poor.

Response: These two sections were updated.

Comment 2: Figure 1 "Detection of phosphoproteins." needs to be improved.

Response: Fig. 1 was improved.

Comment 3: Table 1 “Disease marker and number of samples in HCC” needs to be improved.

Response: Table 1 has been improved in the revised manuscript.

Comment 4: Indicate the element of novelty.

Response: We added a novelty element at the end of abstract and introduction.

Reviewer 2 Report

The reviewed paper is a review paper in the field of using phosphorylated proteins as a potential diagnostic biomarker of cancer. Paper is very interesting and easy to follow, with significant state-of-the art information provided to the reader.

The language should be additionally checked. There are some parts of the article, where it's unclear whether a singular form should be used or if this is a mistake (e.g. l. 306)

Author Response

We thank the reviewers for their constructive criticisms and evaluations of the manuscript. Their comments and suggestions significantly contributed to strengthen this revised manuscript. In the revised manuscript, every effort has been made to incorporate all suggestions made by the reviewers, and address their concerns. All changes/updates in the manuscript were underlined. 

"The reviewed paper is a review paper in the field of using phosphorylated proteins as a potential diagnostic biomarker of cancer. Paper is very interesting and easy to follow, with significant state-of-the art information provided to the reader."

Response: We thank the reviewer for finding our manuscript suitable for the journal.

Comment 1: The language should be additionally checked. There are some parts of the article, where it's unclear whether a singular form should be used or if this is a mistake (e.g. l. 306).

Response: The whole revised manuscript has now been read by English language expert and edited.

Reviewer 3 Report

This paper provides a concise and well-organized summary of the utility of phosphoproteins in cancer management, and may be useful as an introduction for audience unfamiliar with this field. I have some comments to improve the manuscript.

1.      Line 159: ROC should be defined out at its first appearance.

2.      Line 167: “adeno lung carcinoma” should be “adenocarcinoma.”

3.      Lines 141, 185, 186: Locations of comma in numbers, such as 3,70,000 which should be 370,000, should be corrected.

4.      Lines 190-191: This sentence should be revised, because I cannot understand what the authors want to say. In addition, citation of reference 35 may not be appropriate judging from the content of the reference.

5.      Line 220: (2017) should be (42).

6.      Lines 242-243: “One lakh” is not appropriate for audience other than the Indian. It should be replaced with “one hundred thousand.”

7.      Line 251: Period should be placed after “healthy individual.”

8.      protein.”Line 263: “few serum protein” should be “a few serous protein.”

9.      Line 276: “Kidney cancer” should be deleted.

10.  Line 286: “conducted to” should be “was conducted to.”

11.  Line 310: “but” should be deleted.

12.  Table 3: Serum biomarkers of various cancers should be reconsidered. For example, cytokeratin 19 is not a marker of HCC but a marker of cholangiocarcinoma. HCC rather express cytokeratin 18. CA125 is not usually used as a serum marker of gall bladder cancer. CA125 is usually used as a marker of ovarian cancer. CEA and CA19-9 are usually used as a serum marker of gallbladder cancer, cholangiocarcinoma, and pancreatic cancer. CA15-3 is used as a marker of breast cancer.

Author Response

We thank the reviewers for their constructive criticisms and evaluations of the manuscript. Their comments and suggestions significantly contributed to strengthen this revised manuscript. In the revised manuscript, every effort has been made to incorporate all suggestions made by the reviewers, and address their concerns. All changes/updates in the manuscript were underlined. 

"This paper provides a concise and well-organized summary of the utility of phosphoproteins in cancer management, and may be useful as an introduction for audience unfamiliar with this field. I have some comments to improve the manuscript."

Response: We thank the reviewer for finding our manuscript suitable for the journal.

 Comment 1: Line 159: ROC should be defined out at its first appearance.

Response: ROC has been defined.

Comment 2: Line 167: “adeno lung carcinoma” should be “adenocarcinoma.”

Response: Corrected as advised.

Comment 3: Lines 141, 185, 186: Locations of comma in numbers, such as 3,70,000 which should be 370,000, should be corrected.

Response: Corrected

Comment 4: Lines 190-191: This sentence should be revised, because I cannot understand what the authors want to say. In addition, citation of reference 35 may not be appropriate judging from the content of the reference.

Response: This has been revised to bring more clarity.

Comment 5: Line 220: (2017) should be (42).

Response: Corrected

Comment 6: Lines 242-243: “One lakh” is not appropriate for audience other than the Indian. It should be replaced with “one hundred thousand.”

Response: One lakh is replaced with one hundred thousand.

Comment 7: Line 251: Period should be placed after “healthy individual.”

Response. Corrected

Comment 8: protein.”Line 263: “few serum protein” should be “a few serous protein.”

Response: Corrected

Comment 9: Line 276: “Kidney cancer” should be deleted.

Response: Deleted

Comment 10: Line 286: “conducted to” should be “was conducted to.”

Response: Corrected

Comment 11: Line 310: “but” should be deleted.

Response: Deleted

Comment 12: Table 3: Serum biomarkers of various cancers should be reconsidered. For example, cytokeratin 19 is not a marker of HCC but a marker of cholangiocarcinoma. HCC rather express cytokeratin 18. CA125 is not usually used as a serum marker of gall bladder cancer. CA125 is usually used as a marker of ovarian cancer. CEA and CA19-9 are usually used as a serum marker of gallbladder cancer, cholangiocarcinoma, and pancreatic cancer. CA15-3 is used as a marker of breast cancer.

Response: We agree with the reviewer that Cytokeratin 19 (CK19) expression displayed the best specificity to cholangiocarcinoma (Lee Chao-Wei et al., 2013; Qin Jian et al., 2018). CK19 is also found in tumor stem cells of HCC and can be a potential biomarker for predicting poor prognosis after surgical and adjuvant therapies (Zhuo Jian-Yong et al., J. Cancer. 2020; 11(17): 5069-5077). However, we deleted CK19 from Table 3 as this is not specific to HCC. Regarding the marker of gallbladder cancer, the reviewer is right and so we replaced CA125 with CA19-9. CA15-3 is added as a marker of breast cancer in Table 3.